# Lithosepermic Acid Restored the Skin Barrier Functions in the Imiquimod-Induced Psoriasis-like Animal Model

**DOI:** 10.3390/ijms23116172

**Published:** 2022-05-31

**Authors:** Li-Ching Chen, Yu-Ping Cheng, Chih-Yi Liu, Jiun-Wen Guo

**Affiliations:** 1Division of Infectious Diseases, Cathay General Hospital, Taipei 10630, Taiwan; leechenglory@gmail.com; 2Department of Dermatology, Cathay General Hospital, Taipei 10630, Taiwan; m0587304.05g@g2.nctu.edu.tw; 3Division of Pathology, Sijhih Cathay General Hospital, New Taipei City 22174, Taiwan; cyl1124@gmail.com; 4Department of Medical Research, Cathay General Hospital, Taipei 10630, Taiwan

**Keywords:** lithospermic acid, skin barrier, inflammatory cytokines, autophagy, psoriasis

## Abstract

(1) Background: Psoriasis is a T helper 1/T helper 17 cells-involved immune-mediated genetic disease. Lithospermic acid, one of the major phenolic acid compounds of Danshen, has antioxidation and anti-inflammation abilities. Due to the inappropriate molecular weight for topical penetration through the stratum corneum, lithospermic acid was loaded into the well-developed microemulsion delivery system for IMQ-induced psoriasis-like dermatitis treatment. (2) Methods: BALB/c mice were administered with topical imiquimod to induce psoriasis-like dermatitis. Skin barrier function, disease severity, histology assessment, autophagy-related protein expression, and skin and spleen cytokine expression were evaluated. (3) Results: The morphology, histopathology, and skin barrier function results showed that 0.1% lithospermic acid treatment ameliorated the IMQ-induced psoriasis-like dermatitis and restored the skin barrier function. The cytokines array results confirmed that 0.1% lithospermic acid treatment inhibited the cutaneous T helper-17/Interleukin-23 axis related cytokines cascades. (4) Conclusions: The results implied that lithospermic acid might represent a possible new therapeutic agent for psoriasis treatment.

## 1. Introduction

Psoriasis is a T helper 1/T helper 17 (Th17) cells-involved immune-mediated genetic disease. Genetic, extrinsic, and intrinsic risk factors are related to the onset of psoriasis. Studies suggest that the key contributors to the pathogenesis are tumor necrosis factor-alpha and Th17/ interleukin-23 cytokine pathways [1,2]. Even though novel biologics have been developed to treat psoriasis, there remains no cure. Nonetheless, psoriasis patients should be treated early in the disease process to minimize physical and psychological harm [2].

*Salvia miltiorrhiza*, the traditional Chinese herbal medicine known as Danshen, has been widely used to improve blood circulation and treat heart diseases. Salvianolic acid B (Sal. B), lithospermic acid (LA, C_27_H_22_O_12_, MW = 538.5), and rosmarinic acid are the major phenolic acid compound of Danshen [3]. A previous study reported that pretreatment of LA attenuates the neurotoxicity and neuro-inflammation effects induced by 1-methyl-4-phenylpyridin in vitro and in vivo [4]. Liu et al. demonstrated that LA inhibits superoxide radicals stimulated by phorbol-12-myristate13-acetate and N-formyl-methionyl-leucyl-phenylalanine in vitro, and also has an anti-inflammatory effect in the gouty arthritis animal model [5]. In addition, Liu et al. reported that LA showed similar anti-inflammatory properties to that of Sal. B among the phenolic acids of Danshen extract [6]. Furthermore, our previous study demonstrated that Sal. B had an anti-psoriasis effect in the imiquimod-induced psoriasis-like animal model [7].

Like Sal. B with unfavorable physicochemical properties, LA also has a molecular weight of over 500 Dalton compound which may hinder its permeation across the stratum corneum [8,9]. An appropriate topical formulation is therefore required for optimal bioavailability and minimal skin irritation [10]. The microemulsion is considered an effective topical delivery system that increases the permeation of active ingredients across the skin barrier for dermatological treatment or cosmeceutical purposes. It is defined as a clear, transparent, thermodynamically stable mixture of oil, water, and surfactants that encapsulate emulsion droplets of micron to nanometer-sized particles [10,11,12]. As such, our previous well-developed topical microemulsion delivery system [7] was used in this study to investigate the therapeutic effects and its underlying mechanisms in the imiquimod (IMQ)-induced psoriasis-like animal model.

## 2. Results

### 2.1. 0.1% LA and DXM Improved Psoriasis-like Dermatitis

To clarify the effects of LA on psoriasis-like dermatitis, the morphology, histopathology, skin barrier function, and the proliferation of keratinocytes were examined first. The morphology of all groups showed scaly, erythematous, and dry skin conditions (Figure 1A). The histopathology results showed acanthosis, parakeratosis, tortuous capillary dilatation in the papillary dermis, and inflammatory cell infiltration in all groups (Figure 1B). In contrast, the 0.1% LA and Esperson treatment groups showed fewer severe clinical and pathological features than any of the other groups (Figure 1A,B). The results indicate that 0.1% LA and DXM treatment ameliorated the IMQ-induced psoriasis-like dermatitis.

### 2.2. LA and DXM Restored Barrier Function

The 0.1% LA (20.36 ± 2.84 g/m^2^/h) and Esperson (22.29 ± 4.53 g/m^2^/h) treatment significantly restored the transepidermal water loss (TEWL) values compared to those of the control (35.90 ± 2.78 g/m^2^/h) and vehicle (32.31 ± 3.90 g/m^2^/h) groups (all *p* < 0.01, Figure 1C). The 0.025% LA (27.35 ± 2.84 g/m^2^/h) treatment also recovered TEWL values compared to the control group (*p* < 0.05, Figure 1C). The 0.1% LA group (25.34 ± 7.69 arbitrary units (AU)) showed higher skin hydration values than that of the control group (6.33 ± 1.58 AU), vehicle (10.84 ± 2.71 AU) and Esperson (14.68 ± 2.79 AU) groups (all *p* < 0.01, Figure 1D), while 0.025% LA (13.34 ± 3.12 AU) and Esperson treatment showed higher skin hydration values than that of the control group (both *p* < 0.01, Figure 1D). Both 0.1% LA (27.64 ± 3.93 AU) and Esperson (28.69 ± 3.93 AU) treatment decreased the erythema values compared to the control (35.21 ± 3.86 AU) and vehicle (34.95 ± 1.92 AU) (all *p* < 0.01, Figure 1E) groups. Meanwhile, the normal untreated TEWL, skin hydration, and erythema values are 7.38 ± 0.69 g/m2/h, 57.51 ± 2.84 AU, and 14.49 ± 2.49 AU, respectively. The skin barrier measurement results indicate that 0.1% LA and DXM treatment restored the IMQ-induced barrier disruptions.

### 2.3. 0.1% LA and DXM Inhibited Hyperproliferation of Keratinocytes

The ELISA results showed that the expression of proliferation cell nuclear antigen (PCNA), a marker strictly associated with cell proliferation, was significantly decreased in the 0.1% LA (*p* < 0.01) and Esperson (*p* < 0.01) treatment groups (Figure 1F) compared to the control and vehicle groups. The result indicates that 0.1% LA and DXM treatment ameliorated the IMQ-induced hyperproliferation of keratinocytes.

### 2.4. LA Showed a Dose-Dependent Upregulation of Autophagy-Related Protein Expression in IMQ-Induced Autophagy Dysfunction

Two well-characterized autophagy biomarker proteins, LC3B and p62 were selected to examine the effect of LA on autophagy function. LA treatment showed a dose-dependent upregulation of light chain 3B (LC3B) and p62 protein expression compared to the control group, however, this increase did not reach statistical significance (*p* > 0.05; Figure 2A,B). In contrast, Esperson treatment significantly increased LC3B protein expression compared to the control, vehicle, and 0.025% LA treatment groups (all *p* < 0.01). However, the Esperson treatment does not affect the protein expression of P62 (Figure 2A,B). The autophagy-related protein expression result indicates that DXM treatment ameliorated the IMQ-induced autophagy dysfunction.

### 2.5. 0.1% LA and DXM Inhibited IL-17A, IL-17F, IL-22, IL-23, and IL-6 Protein Expression in Psoriasis-like Skin

Th-17/and IL-23 axis-related cytokines were examined to clarify the molecular mechanism of LA in IMQ-induced psoriasis-like dermatitis. The cytokine protein expression results showed that both 0.1% LA and DXM treatment inhibited IL-17A, IL-17F, IL-22, IL-23, and IL-6 protein expression compared to the control group (all *p* < 0.01, Figure 3A–E) and vehicle group (all *p* < 0.05, Figure 3A–E). While 0.025% LA treatment inhibited IL-17A (*p* < 0.05, Figure 3A) and IL-6 (*p* < 0.05, Figure 3E) protein expression compared to the control group. However, no treatment affected the protein expression of TNF-α (*p* > 0.05, Figure 3F).

### 2.6. DXM Inhibited Spleen Enlargement in IMQ-Induced Psoriasis-like Mice

To clarify the systemic effect of LA in IMQ-induced psoriasis-like dermatitis, the spleen weight and Th-17/IL-23 axis-related cytokines from the spleen were examined. The topical treatment of LA did not reduce the enlargement of the spleen (Figure 4A) or spleen weight (*p* > 0.05; Figure 4B) after six consecutive days of IMQ application. However, Esperson treatment significantly reduced the enlargement of the spleen (Figure 4A) and reduced the spleen weight compared to the control, vehicle, and both LA treatment groups (*p* < 0.01; Figure 4B).

### 2.7. DXM Inhibited IL-22, IL-6, IFN-γ, and TNF-α Protein Expression of Spleen in IMQ-Induced Psoriasis-like Mice

The cytokine protein expression results showed that six consecutive days of topical IMQ dramatically raised the Th-17/IL-23 axis-related protein expression in the spleen (all *p* < 0.05; Figure 5A–E) but not TNF-α (*p* > 0.05; Figure 5F) compared to an untreated group of normal health. In contrast, DXM treatment inhibited the IL-22, IL-6, IFN-γ, and TNF-α protein expressions compared to the control group (all *p* < 0.05, Figure 5C–F) and vehicle group (all *p* < 0.05, Figure 5C–F). In contrast, LA treatment did not affect the Th-17/IL-23 axis-related protein expression in the spleen (all *p* > 0.05; Figure 5A–F).

### 2.8. 0.1% LA Inhibited Tape-Stripping (TP) Induced Skin Erythema

To clarify the effect of LA on the skin barrier recovery, acute skin barrier damage induced by tape stripping (TP) study was conducted. 0.1% LA treatment showed the peak TEWL value (65.43 ± 3.92 g/m^2^/h) at 2 h, while the TP control group showed 64.45 ± 5.28 g/m^2^/h at 1 h (Figure 6A). Furthermore, 0.1% LA treatment showed about 25% (23.92 ± 15.99%) of the mean barrier recovery rate at 24 h, while the TP control showed that of 12% (12.01 ± 17.10%). In contrast, 0.1% LA treatment dramatically inhibited TP-induced skin erythema. The skin erythema value returned to normal baseline (−1 h; 14.85 ± 2.09 AU) at 24 h (16.39 ± 1.57 AU) (Figure 6B). This data implies that LA restores barrier function by inhibiting skin erythema.

## 3. Discussion

LA is one of the major phenolic acid compounds of Danshen [3] and showed similar anti-inflammatory properties to that of Sal. B, another phenolic acid compound of Danshen extract [6]. Because of the inappropriate molecular weight for topical penetration through the stratum corneum, LA was loaded in the well-developed microemulsion delivery system for IMQ-induced psoriasis-like dermatitis treatment. The morphology, histopathology, and skin barrier function results showed that 0.1% LA treatment ameliorated the IMQ-induced psoriasis-like dermatitis and restored the skin barrier function. The therapeutic effect is similar, if not less, to the Esperson (0.25% desoximetasone ointment), the positive control, which is one of the first-in-line corticosteroid ointment choices for dermatologists. The cytokines array results confirmed that 0.1% LA treatment inhibited the cutaneous Th-17/IL-23 axis-related cytokines cascades. Together, it implied LA could be a potential therapeutic agent for inflammatory skin diseases.

Autophagy is essential for skin homeostasis. The dysfunctional regulation of autophagy responses leads to several skin disorders, including psoriasis [13,14]. LC3 and p62 are two well-characterized autophagy biomarkers used to represent and quantify the status of autophagy [13,14,15]. Previous studies reported that LC3B protein expression was lower in psoriatic lesional skin than in non-lesional skin [16,17]. In comparison, p62 showed a higher mRNA expression in vitro [18] and a higher protein expression detected by western blot in vivo [19,20]. Our results confirmed decreased LC3B protein expression in IMQ-induced psoriasis-like lesioned skin, compared to normal non-treated skin. Unlike the previous reports, our ELISA results showed a downregulation repression pattern of p62 in psoriasis-like lesioned skin. However, the LC3B and p62 protein expression was upregulated by applying 0.25% desoximetasone ointment. Of importance, Hill et al. pointed out that autophagy is a dynamic process [13]. Additionally, Guo et al. reported higher levels of LC3B and lower levels of P62 of autophagic activity by using an ELISA kit in urticarial lesion rats [15]. Therefore, the sampling time, detected method or the sensitivity, and quality of ELISA kits may influence the autophagy-related protein, such as p62 expression results. On the other hand, our results showed that 0.1% LA was involved in regulating the local cutaneous Th-17/IL-23 axis related cytokines response; however, 0.1% LA treatment only showed a promoted response but without the statistical significance of the LC3B and P62 protein expression compared to the control group. This may imply that LA may involve cutaneous autophagy, but further study, by applying a higher dose of LA treatment, is needed.

Psoriasis is a chronic, systemic immune-mediated disease involving an intricate pathogenic interaction between the innate and adaptive immune systems [21,22]. New treatment or management strategies to reduce systemic inflammation in patients with moderate-to-severe psoriasis may reduce the risk for comorbidities such as cardiovascular disease, diabetes mellitus, obesity, inflammatory bowel disease, and nonalcoholic fatty liver disease [21]. Our results showed that spleen weight and cytokine levels, including IL-17A, IL-22, IL-23, IL-6, and IFN-γ, increased in the IMQ-induced psoriasis-like dermatitis mice. However, spleen weight, IL-6, and IFN-γ showed only a decrease in 0.25% desoximetasone ointment treatment, but not in the LA treatment group. The molecular weight of desoximetasone is 376.5 dalton. It is smaller than lithospermic acid (MW 538.5). Based on the 500 Dalton rule, desoximetasone is easier to penetrate the stratum corneum than lithospermic acid. Furthermore, lithospermic acid needs an appropriate topical formulation to enhance its permeation across the stratum corneum and increase its therapeutic effect. This may imply that only a tiny amount of desoximetasone or LA penetrated the stratum corneum and then into blood circulation. Thus, the limited systemic circulation amount of desoximetasone or LA may not be able to regulate the systemic Th-17/IL-23 axis cytokines.

On the other hand, IL-17F was not detected in all spleen samples, while IFN-γ was not detected in all skin samples. This may be due to the amount of protein expression and the sensitivity of the cytokines array assay kit although, the topical LA treatment showed no effect on the systemic Th-17/IL-23 axis cytokines expression. The 0.1% LA treatment inhibited the cutaneous Th-17/IL-23 axis-related cytokines cascades. Therefore, to clarify the effect of LA on systemic inflammation, an oral administration study is needed for further research.

Erythema, defined as the redness of the skin, results from injury or irritation induced by superficial capillaries dilatation [23,24]. Erythema is also an important indicator of clinical quantification in the psoriasis area, and the severity index score. A previous study reported that thermal imaging may be a useful tool for quantifying inflammation [25]. Our results showed that the 0.1% LA treatment significantly inhibited the Th-17/IL-23 axis-related cytokines expression and also suppressed the skin erythema in the IMQ-induced psoriasis-like study. This may confirm that the erythema value may imply and represent the localized skin inflammation condition. In the TP-induced acute skin barrier disruption study, the results showed that 0.1% LA dramatically restrained the skin erythema and delayed the occurrence of peak transepidermal water loss. This result may imply that LA restores barrier function mainly through inhibiting the localized inflammation response and partially regulating the autophagy process.

## 4. Materials and Methods

### 4.1. Microemulsion Preparation

The formulations were prepared as in our previous study [7].

### 4.2. Animals

Forty-eight male BALB/c mice (7–8 weeks old, BioLASCO Taiwan Co. Ltd., Taipei City, Taiwan) were housed under the standard laboratory conditions of controlled humidity (40%) and temperature (24 ± 2 °C), with a 12 h light/dark cycle. All animal experiments were conducted according to accepted standards of humane animal care, under protocols approved by the Institutional Animal Care and Use Committee (IACUC) of Cathay General Hospital (IACUC110-025, 5 July 2021). During the experimental period, each animal was housed in a separate cage to prevent activities between mice that could affect the measurements of skin barrier functions. In addition, each individual cage was provided with wooden bars to comply with IACUC regulations.

### 4.3. Imiquimod (IMQ)-Induced Psoriasis-like Skin Animal Model

Psoriasiform dermatitis was induced in the mice following our previous study [7]. Thirty-six mice were equally and randomly assigned to either the (a) normal group: healthy without any treatment; (b) control group: IMQ-induced only; (c) vehicle group: IMQ-induced plus vehicle treatment; (d) 0.025% LA group: IMQ-induced plus 0.025% of LA (purity > 98%; Wuhan ChemFaces Biochemical CO., Ltd., Wuhan, Hubei, China) in vehicle treatment; (e) 0.1% LA group: IMQ-induced plus 0.1% of LA in vehicle treatment; or (f) Esperson ointment group: IMQ-induced plus Esperson (0.25% desoximetasone ointment (DXM) Sanofi, Handok Inc., Seoul, Korea) treatment as a positive control. After 3–4 h of IMQ application, 100 μL of drug, vehicle, or 60 mg Esperson was applied once daily to the dorsal skin.

### 4.4. Assessment of Barrier Function

Transepidermal water loss (TEWL), skin hydration, and skin erythema values were measured on the dorsal surface of the mice on day 6 using an MPA 2 system equipped with Tewameter TM300, Corneometer CM825, and Mexmeter MX18 probes (Courage and Khazaka, Köln, Germany).

### 4.5. Collection of Skin and Spleen Specimens

The mice were sacrificed on day 6 after the barrier function assessment. The full-thickness mouse skin was separated into two samples for histological staining and proteins were extracted. The spleen was weighed and then stored at −80 °C before analysis.

### 4.6. Autophagy Cytokine Protein Determination

Proteins were extracted from the epidermis for the autophagy protein analysis p62, and LC3B. Protein samples were then stored at −80 °C before analysis. P62, and LC3B protein expression were analyzed using the p62 Elisa kit (Enzo Life Science, Inc., Farmingdale, NY, USA) and mouse MAP1LC3B Elisa kit (Aviva Systems Biology Corporation, San Diego, CA, USA), respectively. Samples were then determined by a microplate reader (BioTek, Winooski, VT, USA) set to 450 nm wavelengths according to the manufacturer’s instructions.

### 4.7. Inflammatory Cytokine Protein Determination

Proteins were extracted from the whole skin and spleen for the cytokine analysis interleukin-17A (IL-17A), interleukin-17F (IL-17F), interleukin-22 (IL-22), interleukin-23 (IL-23), interleukin-6 (IL-6), Interferon gamma (IFN-γ), and tumor necrosis factor-alpha (TNF-α)). Protein samples were stored at −80 °C before analysis. IL23 protein expression was then analyzed using the mouse IL23 kit (BioLegend, San Diego, CA, USA) and determined by a microplate reader (BioTek, Winooski, VT, USA) set to 450 nm wavelengths according to the manufacturer’s instructions. The other cytokines were determined by the LEGENDplex Kits (mouse Th cytokine panel, BioLegend, San Diego, CA, USA). The samples were incubated with labeled microbeads and each cytokine was determined by flow cytometry (Accuri C6, BD Biosciences, San Jose, CA, USA) according to the manufacturer’s instructions.

### 4.8. Proliferation Cell Nuclear Antigen (PCNA) Protein Determination

Epidermal protein was extracted from the epidermis prepared from full-thickness mouse skin and the PCNA concentrations were determined by a mouse PCNA ELISA quantization kit (Cell Biolabs Inc., San Diego, CA, USA) according to the manufacturer’s instructions.

### 4.9. Histological Staining

Skin samples were fixed in 10% formalin. After routine processing and embedding in paraffin, 5 μm-thick sample sections were cut and then stained with hematoxylin and eosin. The slides were scanned with a slide scanner (3DHISTECH, Budapest, Hungary) at a magnification of ×200.

### 4.10. Barrier Recovery Study

Mice skin barrier disruption was induced by tape stripping as described in previous studies [26]. While the transepidermal water loss value reached 60 ± 10 g/m^2^/h, twelve mice were equally and randomly assigned to either the (a) control: tape stripping only, or (b) 0.1% LA group: tape stripping plus 0.1% LA treatment. After disrupting the barrier, 100 μL of LA was applied immediately to the dorsal skin. The barrier recovery was then monitored by the MPA 2 system equipped with Tewameter TM300, and Mexmeter MX18 probes (Courage and Khazaka, Köln, Germany) at 0, 1, 2, 3, and 24 h.

### 4.11. Statistical Analysis

Data were presented as mean ± standard deviation. One-way ANOVA was followed by the Schffe post-hoc test to determine statistical significance. Student *t*-test was used to compare transepidermal water loss and the skin erythema value in this barrier recovery study. IBM SPSS Statistics Software version 20 (IBM, Armonk, NY, USA) was used. *p* < 0.05 was considered statistically significant.

## 5. Conclusions

Loaded into a well-developed topical delivery system, LA restored the skin barrier function and improved the therapeutic effect by inhibiting the IL-17/IL-23 axis cytokines protein which may also regulate autophagy-related protein expression in the IMQ-induced psoriasis-like dermatitis in mice. These results implied that LA may be a possible new therapeutic agent for psoriasis treatment.

## Figures and Tables

**Figure 1 ijms-23-06172-f001:**
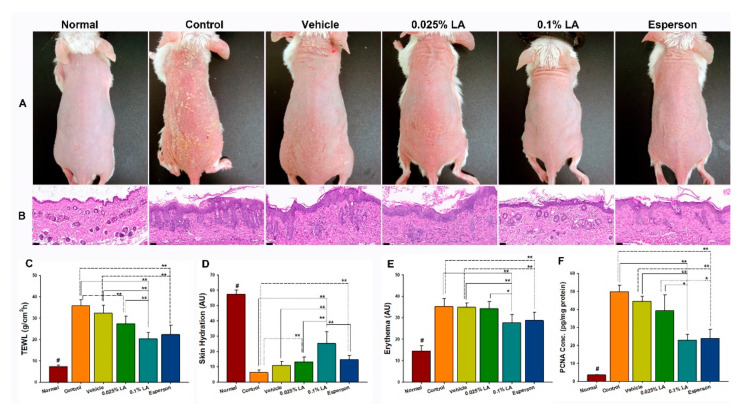
0.1% LA and DXM improved Psoriasis-like Dermatitis. (**A**) Morphology results show that 0.1% LA and Esperson treatment groups have fewer scaly, erythematous, and dry skin features than other groups. (**B**) The H&E stain results show less acanthosis, parakeratosis, tortuous capillary dilatation in the papillary dermis, and inflammatory cell infiltration in the 0.1% LA and Esperson treatment groups. Scale bar = 50 μm. (**C**) Transepidermal Water Loss (TEWL), (**D**) Skin hydration, and (**E**) Erythema values show that 0.1% LA and Esperson treatment groups restored barrier function. (**F**) 0.1% LA and DXM treatment inhibited PCNA protein expression. LA, lithospermic acid; PCNA, proliferation cell nuclear antigen; # *p* < 0.01, compared to all the other groups; * *p* < 0.05; ** *p* < 0.01.

**Figure 2 ijms-23-06172-f002:**
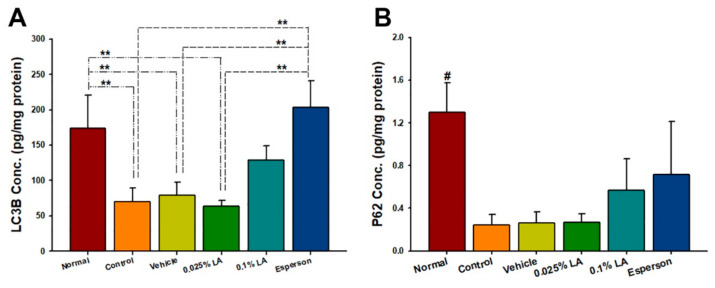
LA showed a dose-dependent upregulation of autophagy-related proteins. (**A**) LC3B (**B**) P62 protein expression. LA, lithospermic acid; # *p* < 0.01, compared to all the other groups; ** *p* < 0.01.

**Figure 3 ijms-23-06172-f003:**
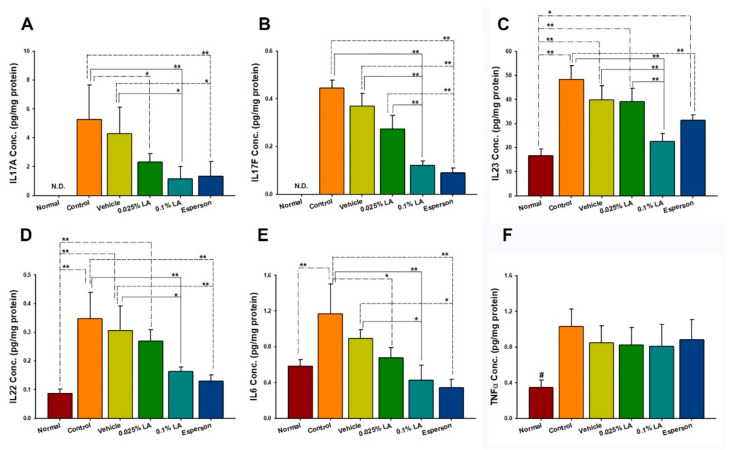
0.1% LA and DXM Inhibited (**A**) IL-17A, (**B**) IL-17F, (**C**) IL-23, (**D**) IL-23, and (**E**) IL-6 but not (**F**) TNF-α Protein Expression in Psoriasis-like Skin. IL, interleukin; # *p* < 0.01, compared to all the other groups; * *p* < 0.05; ** *p* < 0.01.

**Figure 4 ijms-23-06172-f004:**
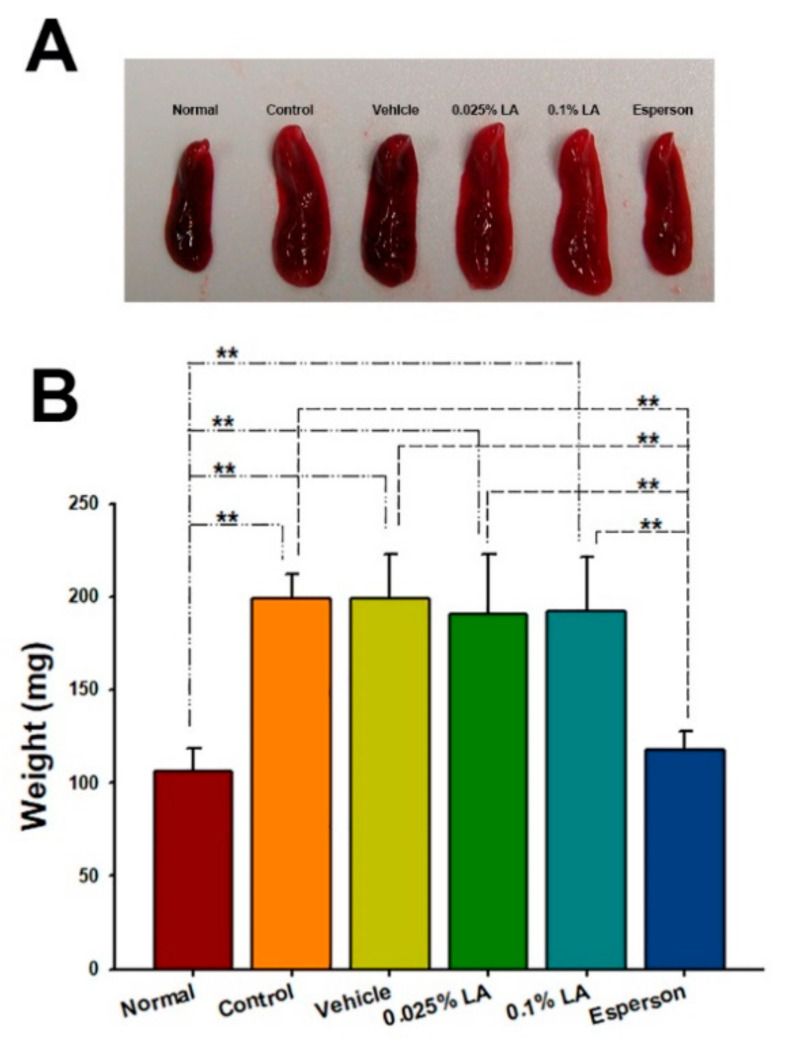
DXM inhibited spleen enlargement in IMQ-induced psoriasis-like mice. (**A**) Morphology (**B**) Measurement of spleen weight. ** *p* < 0.01.

**Figure 5 ijms-23-06172-f005:**
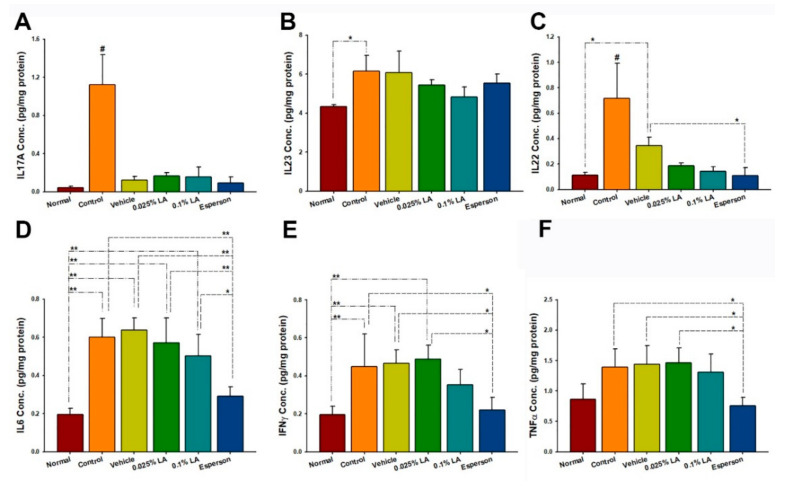
LA did not regulate (**A**) IL-17A, (**B**) IL-23, (**C**) IL-22, (**D**) IL6, (**E**) IFN-γ, and (**F**) TNF-α protein expression in the spleen of IMQ-induced psoriasis-like dermatitis. IL, interleukin; # *p* < 0.01, compared to all the other groups; * *p* < 0.05; ** *p* < 0.01.

**Figure 6 ijms-23-06172-f006:**
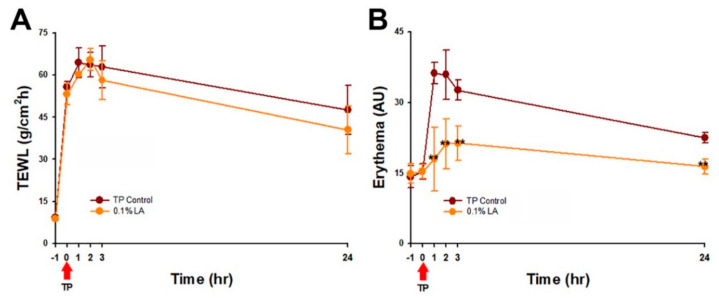
0.1% LA inhibited tape-stripping induced skin erythema. (**A**) TEWL, and (**B**) skin erythema recovery curve after tape stripping. The red arrow indicated the start of tape-stripping. TP, tape-stripping; LA, lithospermic acid, hr, hour; ** *p* < 0.01.

## Data Availability

The datasets used and/or analyzed during the current study are available from the corresponding author on reasonable request.

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
