# Peer review of "Lithosepermic Acid Restored the Skin Barrier Functions in the Imiquimod-Induced Psoriasis-like Animal Model"

_ijms, 2022, doi:10.3390/ijms23116172_

Round 1
Reviewer 1 Report
Chen et al. examined the beneficial effects of topically applied lithospermic acid (LA), a phenolic compound in Danshen.
Using the imiquimod (IMQ)-induced psoriasis-like skin inflammation model, they showed LA improved major symptoms without overt effects on the up-regulation of inflammatory markers, claiming that LA’s therapeutic potential for psoriasis.
The reviewer raised some concerns about improving the clarity of the manuscript.
Major concerns.
1. “L165: The therapeutic effect is similar if not less to the Esperson ~.”
This reviewer agrees that LA improves skin barrier function, at least based on Fig. 1C–F. The potentially interesting points of the study could be the discrepancy between the epidermal barrier function and the systemic inflammatory responses (splenomegaly or cytokine expression). The former is comparable with the latter inferior to the dexamethasone (DEX). The reviewer asks the authors to describe the underlying mechanisms causing this discrepancy, i.e., is the DEX’s general immunosuppressive effects or relatively low molecular weight making it easier to penetrate the stratum corneum (SC) barrier?
2. It seems that LA does not have major effects on autophagy, as determined by the effectors LC3B/p62. This study is supposed to examine the effects of LA, but not DEX. If the authors want to show the results for legitimate reasons, please describe them explicitly. Otherwise, the data might cause confusion and impairs the readability.
3. In the discussion part contains a lot of limitation statements. If any of these have already been tested, they should present the data; LA oral gavage or high-dose LA application may have yielded very different outcomes.
4. Alternatively, because 0.1% LA looks promising for the recovery of epidermal barrier function, why the authors should at least test the transepidermal water recovery rates following the removal of SC by cellophane tape, which is a standardized method and has good readouts (refer to such as PMID: 23223142).
Minor concerns.
1. L36 acid were, rather than was
2. Please describe the microemulsion delivery system briefly so that the readers can comprehend the advantage of this drug delivery system.
3. Fig. 1B requires higher magnification pictures (such as 100x).
Author Response
Reviewer 1
Chen et al. examined the beneficial effects of topically applied lithospermic acid (LA), a phenolic compound in Danshen.
Using the imiquimod (IMQ)-induced psoriasis-like skin inflammation model, they showed LA improved major symptoms without overt effects on the up-regulation of inflammatory markers, claiming that LA’s therapeutic potential for psoriasis.
The reviewer raised some concerns about improving the clarity of the manuscript.
Major concerns.
- “L165: The therapeutic effect is similar if not less to the Esperson ~.”
This reviewer agrees that LA improves skin barrier function, at least based on Fig. 1C–F. The potentially interesting points of the study could be the discrepancy between the epidermal barrier function and the systemic inflammatory responses (splenomegaly or cytokine expression). The former is comparable with the latter inferior to the dexamethasone (DEX). The reviewer asks the authors to describe the underlying mechanisms causing this discrepancy, i.e., is the DEX’s general immunosuppressive effects or relatively low molecular weight making it easier to penetrate the stratum corneum (SC) barrier?
Response: The molecular weight of desoximetasone is 376.5. It is smaller than that of lithospermic acid (MW 538.5). Based on the 500 Dalton rule, desoximetasone is easier to penetrate the stratum corneum than that lithospermic acid. Also, lithospermic acid needs an appropriate topical formulation to enhance its permeation across the stratum corneum and then increase its therapeutic effect. This may imply that only a tiny amount of desoximetasone or LA penetrated the stratum corneum and then into blood circulation. Thus, the limited systemic circulation amount of desoximetasone or LA may not be able to regulate the systemic Th-17/IL-23 axis cytokines. We revised the discussion section in the manuscript (page 7, Line 221-226).
- It seems that LA does not have major effects on autophagy, as determined by the effectors LC3B/p62. This study is supposed to examine the effects of LA, but not DEX. If the authors want to show the results for legitimate reasons, please describe them explicitly. Otherwise, the data might cause confusion and impairs the readability.
Response: We appreciate your comments. We revised the section title to “LA showed a dose-dependent upregulation effect on the autophagy-related protein expression in IMQ-induced autophagy dysfunction” We also revised the section text. (page 3-4, Line 105-117)
- In the discussion part contains a lot of limitation statements. If any of these have already been tested, they should present the data; LA oral gavage or high-dose LA application may have yielded very different outcomes.
Response: We did not test the oral and high dose LA application study yet. Because by application the 0.1% LA, the data showed the improved therapeutic effect. Thus, we conduct the tape stripping study as you suggest.
- Alternatively, because 0.1% LA looks promising for the recovery of epidermal barrier function, why the authors should at least test the transepidermal water recovery rates following the removal of SC by cellophane tape, which is a standardized method and has good readouts (refer to such as PMID: 23223142).
Response: We appreciate your comments. The 0.1% LA treatment showed a significantly inhibition effect on the skin erythema values induced by tape stripping. In contrast, 0.1% LA showed a fast but not significantly barrier recovery after tape stripping. We revised the method(page 9, Line 313-320), results(page 6, Line 165-178), and discussion sections(page 8, Line 237-248).
Minor concerns.
- L36 acid were, rather than was
Response: Thank you for your suggestions. We revised this typing error. (page 1, Line 36)
- Please describe the microemulsion delivery system briefly so that the readers can comprehend the advantage of this drug delivery system.
Response: Thank you for your suggestions. We add a briefly note on microemulsion to the introduction section. (page 2, Line 50-54)
- Fig. 1B requires higher magnification pictures (such as 100x).
Response: Thank you for your suggestions. We revised the Fig 1B with higher magnification pictures. (page 2-3)

Reviewer 2 Report
In this paper, authors demonstrated the utility of lithospermic acid, one of the major phenolic acid compounds of Danshen on IMQ-induced psoriasis-like dermatitis. It has antioxidation and anti-inflammation abilities. Due to the inappropriate molecular weight for topical penetration through the stratum corneum, lithospermic acid (LA) was loaded into the well-developed microemulsion delivery system for IMQ-induced psoriasis-like dermatitis treatment. Skin barrier function, disease severity, histology assessment, autophagy-related protein expression, skin and spleen cytokine expression were evaluated. The morphology, histopathology, and skin barrier function results showed that 0.1% LA treatment ameliorated the IMQ-induced psoriasis-like dermatitis and restored the skin barrier function. The cytokines array results confirmed that 0.1% LA treatment inhibited the cutaneous T helper-17/Interleukin-23 axis related cytokines cascades. Authors concluded that LA might represent a possible new therapeutic agent for psoriasis treatment. I have several questions.
1) It is well documented that LA ameliorated the IMQ-induced psoriasis-like dermatitis and restored the skin barrier function. However, the mechanism by which LA restores such dermatitis and skin barrier function is missing. What hypotheses can be considered? Are there any other data, such as in vitro or in vivo data?
2) Similarly, autophagy data have shown that LA improves the IMQ-induced psoriasis-like dermatitis and restored the skin barrier function as a mechanism. Autophagy is a phenomenon found at the cellular level. What are the data on autophagy in the cells for LC3B and P62 obtained in this study? If data are available, what cellular autophagy dysfunctions, if any, are involved in IMQ-induced psoriasis-like dermatitis? In which cells does LA improve psoriasis-like dermatitis by improving autophagy dysfuction?
From the data presented here, I felt that LA is a very interesting substance. It would be a wonderful and promising substance if it could be used not only in mouse models but also in actual clinical practice in the future.
Author Response
Reviewer 2
In this paper, authors demonstrated the utility of lithospermic acid, one of the major phenolic acid compounds of Danshen on IMQ-induced psoriasis-like dermatitis. It has antioxidation and anti-inflammation abilities. Due to the inappropriate molecular weight for topical penetration through the stratum corneum, lithospermic acid (LA) was loaded into the well-developed microemulsion delivery system for IMQ-induced psoriasis-like dermatitis treatment. Skin barrier function, disease severity, histology assessment, autophagy-related protein expression, skin and spleen cytokine expression were evaluated. The morphology, histopathology, and skin barrier function results showed that 0.1% LA treatment ameliorated the IMQ-induced psoriasis-like dermatitis and restored the skin barrier function. The cytokines array results confirmed that 0.1% LA treatment inhibited the cutaneous T helper-17/Interleukin-23 axis related cytokines cascades. Authors concluded that LA might represent a possible new therapeutic agent for psoriasis treatment. I have several questions.
1) It is well documented that LA ameliorated the IMQ-induced psoriasis-like dermatitis and restored the skin barrier function. However, the mechanism by which LA restores such dermatitis and skin barrier function is missing. What hypotheses can be considered? Are there any other data, such as in vitro or in vivo data?
Response: We conduct the tape stripping study as reviewer 1 suggest to clarify the effects/mechanisms of LA on the skin barrier recovery. Thus, we hypotheses that LA restore barrier function mainly through inhibit localized inflammation response and partially regulation autophagy process. We revised the method, results and discussion sections.
2) Similarly, autophagy data have shown that LA improves the IMQ-induced psoriasis-like dermatitis and restored the skin barrier function as a mechanism. Autophagy is a phenomenon found at the cellular level. What are the data on autophagy in the cells for LC3B and P62 obtained in this study? If data are available, what cellular autophagy dysfunctions, if any, are involved in IMQ-induced psoriasis-like dermatitis? In which cells does LA improve psoriasis-like dermatitis by improving autophagy dysfunction?
Response: The samples for autophagy-related protein analysis were epidermis which mainly are the keratinocytes. We collected and lysed the epidermis. Thus, it is difficult to answer what cellular autophagy dysfunctions. However, our methods and results could demonstrate that LA improve psoriasis-like dermatitis by improving autophagy dysfunction in keratinocytes.
From the data presented here, I felt that LA is a very interesting substance. It would be a wonderful and promising substance if it could be used not only in mouse models but also in actual clinical practice in the future.
Response: We appreciate your comments.

Round 2
Reviewer 1 Report
No comments.